# Plasmatic Inactive IL-18 Predicts a Worse Overall Survival for Advanced Non-Small-Cell Lung Cancer with Early Metabolic Progression after Immunotherapy Initiation

**DOI:** 10.3390/cancers16122226

**Published:** 2024-06-14

**Authors:** Serena Janho dit Hreich, Olivier Humbert, Tanguy Pacé-Loscos, Renaud Schiappa, Thierry Juhel, Marius Ilié, Victoria Ferrari, Jonathan Benzaquen, Paul Hofman, Valérie Vouret-Craviari

**Affiliations:** 1University Côte d’Azur, CNRS, INSERM, Institute for Research on Cancer and Aging (IRCAN), Team 4, 06108 Nice, France; serenahreich@gmail.com (S.J.d.H.); thierry.juhel@univ-cotedazur.fr (T.J.); ilie.m@chu-nice.fr (M.I.); hofman.p@chu-nice.fr (P.H.); 2FHU OncoAge, 06108 Nice, France; 3Department of Nuclear Medicine, Centre Antoine Lacassagne, 06100 Nice, France; olivier.humbert@nice.unicancer.fr; 4University Côte d’Azur, CNRS, INSERM, Institut Biologie Valorse, Team Humbert, 06108 Nice, France; 5Department of Epidemiology, Biostatistics and Health Data, Centre Antoine Lacassagne, 06100 Nice, France; tanguy.pace-loscos@nice.unicancer.fr (T.P.-L.); renaud.schiappa@nice.unicancer.fr (R.S.); 6IHU RespirERA, Pasteur Hospital, 06000 Nice, France; 7Laboratory of Clinical and Experimental Pathology, Hospital-Related Biobank (BB-0033-00025), Pasteur Hospital, 06000 Nice, France; 8Department of Medical Oncology, Centre Antoine Lacassagne, 06100 Nice, France

**Keywords:** immune checkpoint inhibitor, lung adenocarcinoma, IL-18 signaling pathways, neutrophils

## Abstract

**Simple Summary:**

Immune checkpoint inhibitors (ICIs) offer an effective approach to the treatment and potential cure of lung cancer, with unprecedented rates of long-term clinical responses. However, for most patients, early progression is observed after initiation of immunotherapy. In some cases, this is not a true tumor escape but a transient inflammatory response, termed “pseudoprogression”. There are currently no reliable biological or imaging biomarkers that can be used in daily practice and that can accurately distinguish between patients with true progression, for whom treatment should be stopped, and patients with pseudoprogression, for whom treatment should be continued. The aim of our ancillary study was to assess the potential value of plasmatic IL-18 levels (active, free IL-18, and inactive complexed IL-18) as a biomarker of immunotherapy efficacy. While free IL-18 levels tend to decrease in NSCLC patients treated with ICIs, these levels do not serve as a discriminatory factor between responding and nonresponding patients. However, our findings demonstrate that monitoring plasmatic levels of inactive IL-18 provides valuable insights into tracking patient outcomes, particularly in the challenging case of patients with initial tumor progression after ICI initiation.

**Abstract:**

The aim of this study was to assess the potential value of circulating active and inactive IL-18 levels in distinguishing pseudo and true tumor progression among NSCLC patients receiving immune checkpoint inhibitor treatments (ICIs). Methods: This ancillary study includes 195 patients with metastatic non-small-cell lung cancer (NSCLC) treated with ICI in monotherapy, either pembrolizumab or nivolumab. Plasmatic levels of IL-18-related compounds, comprising the inhibitor IL-18 binding protein (IL-18BP), the inactive IL-18 (corresponding to IL-18/IL-18BP complex), and the active free IL-18, were assayed by ELISA. Objective tumoral response was analyzed by ^18^FDG PET-CT at baseline, 7 weeks, and 3 months post treatment induction, using PERCIST criteria. Results: Plasmatic IL-18BP and total IL-18 levels are increased at baseline in NSCLC patients compared with healthy controls, whereas IL-18/IL-18BP complexes are decreased, and free IL-18 levels remain unchanged. Neither of the IL-18-related compounds allowed to discriminate ICI responding to nonresponding patients. However, inactive IL-18 levels allowed to discriminate patients with a first tumor progression, assessed after 7 weeks of treatment, with worse overall survival. In addition, we showed that neutrophil concentration is also a predictive indicator of patients’ outcomes with OS (HR = 2.6, *p* = 0.0001) and PFS (HR = 2.2, *p* = 0.001). Conclusions: Plasmatic levels of inactive IL-18, combined with circulating neutrophil concentrations, can effectively distinguish ICI nonresponding patients with better overall survival (OS), potentially guiding rapid decisions for therapeutic intensification.

## 1. Introduction

The introduction of targeted therapies and immunotherapy, which represent a major advance in the management of non-small-cell lung cancer (NSCLC), profoundly alters treatment paradigms [1]. Although effective, these new approaches come up against the inherent complexity of tumor heterogeneity and interindividual variability in therapeutic response. Targeted therapies, leveraging pharmacological agents designed to address specific molecular alterations in tumor cells, have shown notable efficacy within patient subgroups harboring these alterations [2]. Simultaneously, immunotherapies, primarily distinguished by the deployment of immune checkpoint inhibitors (ICIs), allowing T cells to kill cancer cells by blocking PD-1/PD-L1 signaling, have profoundly revolutionized the management and prognosis of lung cancer devoid of targetable molecular alterations. Despite this transformative impact, the antitumor immune response elicited by these treatments exhibits significant variability among individuals, with around 30% of patients achieving a clinical benefit to ICI, emphasizing the critical need for the development of predictive biomarkers [3]. 

Interleukin-18 (IL-18), formerly referred to as interferon-gamma (IFN-γ) inducing factor [4], interacts with the IL-18Rα (IL-18R1) subunit, leading to the recruitment of the high-affinity IL-18Rβ (IL-18RAP) subunit. This interaction initiates MYD88/NF-κB activation and promotes the transcription of IFN-γ. IFN-γ is recognized for its direct antitumor effects through growth inhibition and its ability to enhance immune cell-mediated killing indirectly, as well as PD-L1 expression [5]. Consequently, IL-18 serves as a pivotal factor in triggering cellular immunity, thus driving the stimulation of an effective antitumor immune response. We therefore hypothesized that IL-18 levels may sensitize tumors to immunotherapy and thus could be a marker of ICI efficacy.

IL-18 exhibits a stronger affinity for its inherent antagonist, known as IL-18 binding protein (IL-18BP). Consistently secreted in substantial quantities, IL-18BP serves as a negative feedback loop to IL-18 activity by actively preventing IL-18 from binding to its receptor. This regulatory mechanism is heightened in response to elevated levels of IFN-γ, contributing to the modulation of IL-18 activity [6].

The contribution of IL-18 to tumor progression is controversial. Preclinical studies have shown that IL-18 is required to inhibit tumor growth using either il18^−/−^ or il18r1^−/−^ mice, or even by administrating recombinant IL-18 [7]. In NSCLC patients, IL-18 has been shown to be expressed by tumor cells and to be active, as suggested by its positive correlation with IFN-γ production [8]. More interestingly, mouse preclinical lung tumor models showed that IL-18 sensitizes lung tumors to anti-PD-1 immunotherapy [9,10]. Coherently, IL-18BP was shown to be elevated after anti-PD-L1 treatment in NSCLC blocking therefore the activity of IL-18 [9,10]. Furthermore, in NSCLC patients treated with atezolizumab, IL-18 showed a dose–exposure relationship that potentially correlates with tumor size [11]. To our knowledge, no previous studies have explored the activity levels of IL-18 in NSCLC by looking at its free unbound form or even its bound inactive one. Consequently, assessing the active status of IL-18 is crucial to understanding its role in tumor progression in patients under immunotherapy.

Treatment efficacy can be measured thanks to Positon Emission Tomography/Computed Tomography (PET/CT), where the standard criteria to evaluate response are the “Positron Emission Tomography Response Criteria in Solid Tumors” (PERCIST) defined by Wahl et al. in 2009 [12]. The criteria consist of four categories: complete metabolic response (CMR), partial metabolic response (PMR), progressive metabolic disease (PMD), and stable metabolic disease (SMD). Our team has previously published results highlighting the relevance of PET/CT in ICI-treated lung cancer. In particular, we showed that ^18^FDG PET enables early assessment of treatment-induced metabolic changes in the tumor or tissue inflammation linked to immune-related adverse events [13]. We also participated in the drafting of guidelines/procedural standards on the recommended use of ^18^FDG PET/CT imaging during immunomodulatory treatments in patients with solid tumors [14].

In this ancillary study, we aimed to determine the plasma levels of free IL-18 (active form) and IL-18/IL-18BP complex (inactive form) before and during immunotherapy treatment in order to determine the predictive value of plasmatic IL-18-related compounds on objective tumor response assessed by ^18^FDG-PET.

## 2. Materials and Methods

### 2.1. Patients

In this study, we collected data from two prospective observational studies conducted within our center (Centre Antoine Lacassagne, Nice) on patients suffering from metastatic NSCLC treated with ICIs alone, regardless of the previous lines of treatment. The patients received one of the following three ICI drugs: pembrolizumab 2 mg/kg every three weeks, nivolumab 240 mg every two weeks, or atezolizumab 1200 mg every three weeks. The first study prospectively included 108 patients from February 2017 to June 2022 (FDG ECMI n°ID-RCB: 2018-A02116-49). The second study prospectively included 87 patients from April 2019 to June 2022 (FDG IMMUN, n°ID-RCB:2018-A00915-50). 

For both studies, inclusion criteria were the following: (1) pathologically proven stage IIIB or IV NSCLC; (2) an indication to start ICI in monotherapy and in the first or later line; (3) ECOG performance status of 0 to 2; (4) age of at least 18 years; (5) histology corresponding to adenocarcinoma, squamous cell carcinoma, or undifferentiated carcinoma; (6) no patient opposition; and (7) no measurable lesion by PERCIST.

These studies were approved by the ethics committees and all patients gave signed and informed consent to participate.

### 2.2. ^18^FDG PET-CT Exams

In each of the two studies, all patients underwent an ^18^FDG-PET/CT within 12 weeks prior to the initiation of treatment (PET baseline), then 7 weeks after the start of treatment (PET_interim_1), and one last one 3 months after the start of treatment (PET_interim_2). 

^18^FDG-PET/CT was performed using two different PET/CT imaging systems: Biograph mCT PET/CT from February 2017 to September 2019 and Biograph Vision 600 PET/CT (Siemens Healthcare, Erlangen, Germany) from September 2019 to January 2023. The patients were asked to fast for at least 6 h before the intravenous injection of 3 MBq/kg (Biograph mCT PET/CT) or 2.5 MBq/kg (Biograph Vision 600 PET/CT) of ^18^FDG.

### 2.3. PERCIST Criteria

The metabolic response was assessed according to the PERCIST criteria. Briefly, a complete metabolic response (CMR) was defined as the disappearance of target lesions uptake and the absence of new lesions, a partial metabolic response (PMR) was defined as a decrease in the sum of the SUV peak of the target lesions (up to 5) by more than 30% in the absence of new lesion, and a stable metabolic disease (SMD) was defined as the absence of criteria to define progression, with either a partial or complete metabolic response. All of these responses define a group of patients termed “metabolic responders” (MR). Finally, in the case of new lesions appearance or progression of target lesions with an increase in the sum of SUV peak by more than 30%, metabolic progression of the disease (PMD) was defined. 

Because of the investigational nature of PET_interim_1 and the knowledge of the specific immune-related response pattern with CT and PET, the result from this scan was not directly used to guide patient management. The treatment was continued even in the case of PMD. Nonetheless, for patient security, the patient’s clinical status was assessed at each course of treatment, and it could be stopped at any time in case of clinical worsening or toxicity. For patients without severe clinical worsening, the metabolic response at 3 months was again evaluated according to the PERCIST criteria (comparing PET_interim_2 to PET_interim_1). However, the pseudoprogression pattern was added and defined as a first PMD at 7 weeks according to PERCIST criteria, followed by a PERCIST response on PET_interim_2. The decision to stop or continue immunotherapy was then made during a multidisciplinary tumor board, confronting the patient’s clinical status and the PET results.

### 2.4. Blood Sample Collection

Human plasma samples were collected at the Centre de Lutte contre le cancer (CLCL Antoine Lacassagne), France, from patients with NSCLC who were treated with ICIs. Two EDTA blood tubes (BD vacutainer, 10 mL, BD, New Jersey, USA) were drawn from each patient. Each tube was centrifuged at 815× *g* for 10 min at 4 °C and the collected supernatant was centrifuged at 2500× *g* for 10 min at 4 °C. Plasma was aliquoted (1ml/aliquot) and stored in the accredited Biobank Cote d’Azur (NFS96-900 [15]) and in the Biological Resource Center of the Centre Antoine Lacassagne (CRB, ISO 9001, NFS96-900 [15]) until their use. Samples from patients with NSCLC were collected sequentially at the time of the first visit to the clinic (baseline) and after PET_interim_ 1 and 2 visits. The complete baseline and treatment characteristics of the patients are summarized in Table 1. This study’s protocols were approved by the Institutional Review Board at the University of Bordeaux (CPP Sud Ouest et Outre mer, approval 08/2019) and all patients provided written informed consent.

### 2.5. IL-18-Related Compound Measurement

Plasma was obtained from citrate-anticoagulated whole blood, which was immediately centrifuged at 1700× *g* for 15 min and stored at −80 °C until further analysis. Human total IL-18 (DY318, R&D Systems, Santa Clara, CA, USA), human IL-18/IL-18 BPa complex (ref DY8936, R&D Systems), and human IL-18 BPa (DY119, R&D Systems) were determined by commercially available ELISA kits according to the manufacturer’s instructions and had minimum detection limits of 11.7 pg/mL, 62.5 pg/mL and 93.8 pg/mL, respectively. The level of free, bioactive IL-18 was calculated based on the mass-action law, using a dissociation constant of 400 pM and a stoichiometric ratio of 1:1 and the following equation was applied: [free IL-18] = (−b+√(b2−4c))/2, with *b* = [IL-18 BP] − [IL-18 total] + Kd, *c* = −Kd × [IL-18 total] and Kd = 400 pM, as previously described [16]. The molecular weight of IL-18 was 18 kDa and IL-18BP 24 kDa.

### 2.6. Microarray

mRNA expression profile was obtained from eTumor Comprehensive Genome Atlas (TCGA) project database. We used TCGA lung adenocarcinoma (LUAD) and squamous cell carcinoma (LUSC) cohort. Microarray was performed on whole-lung homogenate from subjects undergoing thoracic surgery and diagnosed as having NSCLC stage IV as determined by clinical history and surgical pathology. Healthy corresponds to non-tumor-adjacent tissue.

### 2.7. Statistical Analysis

Categorical data were presented as relatives and absolute frequencies. Chi2 test or Fisher test, in the event of noncompliance with the Chi2 application conditions, were used to evaluate the data repartition between groups for categorical variables. Continuous data were presented with mean, median, min, max, and standard deviation. T-test or Wilcoxon rank sum test, in the event of noncompliance with the T-test application conditions, was used to evaluate the distribution similarity of continuous variables. The Shapiro test was used to assess the normality of the continuous variable distribution. Pearson or Spearman coefficient tests, in the event of noncompliance with the Pearson correlation application conditions, were used to measure the dependency between continuous variables. Missing data were presented as absolute numbers and percentages. Data were normalized using the basic logarithmic function. Censored data were presented with the median follow-up (calculated with the inversed Kaplan–Meier method [17] and Kaplan–Meier curve). Survival percentage and 95% confidence interval were presented from 0 to 24 months with 6-month intervals or from 0 to 12 months with 3-month intervals. Survival curves were compared with the log-rank test, and the hazard ratio was calculated from a Cox regression and given with a 95% confidence interval. Overall survival (OS) was calculated between the date of consent and the date of death. Progression-free survival (PFS) was calculated between the date of consent and the date of either tumor progression confirmed by a multidisciplinary tumor board or death. Patients showing no event (death or progression) or lost to follow-up were censored at the date of their last contact. Thresholds for the continuous variable were determined using a brute-force method based on the log-rank test to divide patients into high-risk and low-risk groups based on survival data. Prognostic variables were investigated for OS and PFS using univariate Cox regression. The best multivariable model was selected using a stepwise algorithm using Akaike’s information criterion [18]. The resulting model was used to make a predictive score allowing to distinguish between low- and high-risk groups. Model performance was evaluated using sensitivity and specificity metrics. To correct the inflation of type I error α, in the case of multiple testing, *p*-values were corrected using FDR [19]. A two-sided *p*-value of 0.05 or less was considered significant. All analyses were performed using the R version 4.3.1. Bestcut2 function from the greyzone Surv library was used to find the cut-off for continuous variables.

## 3. Results

### 3.1. Patient Characteristics

Of the two hundred seventy-two patients included in our cohort, 195 patients and 9 healthy controls were evaluated (Appendix A). The mean patient age was 65.8 years. All patients had a locally advanced or metastatic NSCLC, with the pathological type being adenocarcinoma in 77.5%, squamous cell carcinoma in 26%, and other histology type in 5% (Table 1). One hundred four patients (74.8%) were treated with pembrolizumab, 29 received Nivolumab (20.8%) and 6 received Atezolumab (4.3%).

The median time between baseline PET and the introduction of immunotherapy was 9 days [0–61]. The median time between the introduction of immunotherapy and the first intermediary PET (PET1) was 50 days [20–292]. The median time between the introduction of immunotherapy and the second intermediary PET (PET2) was 92 days [71–328]. Patients’ median follow-up was 12.9 months (CI 95%: 12.2–35.1). Patients’ median overall survival (OS) was 17.5 months (CI 95%: 14.4–21.5). Patients’ median progression-free survival (PFS) was 7.9 months (CI 95%: 6.5–9.8) (Appendix A).

### 3.2. Tumor Response on PET_interim_1 (PERSIST Criteria)

The percentage of metabolic responders (MR) according to PERCIST on PET_interim_1 was 40% (44/110) split into 5% with a complete metabolic response (CMR), 23.1% with a partial metabolic response (PMR), and 7.6% with a stable metabolic disease (SMD). Finally, 66% of the patients had a progressive metabolic disease (PMD) at PET_interim1_. Among them, 31 patients (35.6%) were a posteriori confirmed as pseudoprogressors on PET_interim_2 performed 3 months after ICI initiation (metabolic response after an initial PERCIST progression). The other patients had a confirmed progression on PET_interim_2.

### 3.3. Levels of IL-18 Relative Analytes at Baseline in Plasma of NSCLC Patients

We first compared the plasmatic concentration of IL-18-related compounds from NSCLC patients at baseline and healthy controls. Since IL-18 binds to IL-18BP with a higher affinity than IL-18R, the stochiometric ratio being one molecule of IL-18 for one molecule of IL-18BP (Figure 1a), we thought it would be more pertinent to determine the level of free IL-18, which corresponds to the active form of IL-18 [16]. 

We also measured the concentration of IL-18/IL-18BP complex to quantify the inactive IL-18 form. In doing so, we observed that the concentration of plasmatic free IL-18 tends to increase in NSCLC patients, whereas the concentration of IL-18/IL-18BP complex is downregulated (Figure 1b). These results were compared with transcriptomic analyses from publicly available datasets of lung homogenates from stage IV NSCLC patients. The comparison focused on mRNA expression levels of the inflammasome components NLRP3, PYCARD, and CAPS1, which are upstream regulators of mature IL-18 production between control samples and primary solid tumors. We observed a downregulation in the mRNA expression of the constituents of the NLRP3 inflammasome in stage IV NSCLC patients versus controls (Appendix A). We further investigated the signaling pathway downstream of IL-18. We showed that the mRNA expression of IL-18BP, IL-18R1, and IL-18RAP were downregulated in NSCLC patients compared with controls, whereas the levels of IL-18 remained unchanged (Appendix A). Of note, the transcriptomic data did not accurately reflect the total IL-18 and IL-18BP protein levels, which were elevated in the plasma of NSCLC patients (Appendix A). This indicates that intratumoral mRNA content cannot predict the circulating levels of IL-18-related compounds.

We next wondered whether circulating concentrations of IL-18-related compounds may distinguish ICI responders from nonresponders.

### 3.4. The Level of Plasmatic Free IL-18 Is Decreased in ICI-Treated NSCLC Patients

We compared the concentration of each IL-18-related compound at baseline, as well as after 7 weeks (TEP_interim_1) or 3 months (TEP_interim_2) of treatment in all patients and nonresponding or responding patients. We observed that the concentration of free IL-18 progressively decreases during ICI treatment in all patients (Figure 2a, left panel). This effect is lost in the nonresponding group. By contrast, and no matter the studied subpopulations, no significant differences were observed regarding the plasmatic levels of IL-18/IL-18BP complex (Figure 2a, right panel), IL-18BP, or total IL-18 (Appendix A).

Having shown that ICIs impact the plasmatic level of free IL-18, we thought this difference may serve as a biomarker to identify pseudo versus true progressors. Indeed, nonresponding patients at TEP_interim_1 can be further divided into two subpopulations at TEP_interim_2, one corresponding to patients achieving a pseudoprogression (transient tumor progression at TEP 1 followed by tumor response at TEP_interim_2, corresponding to antitumor immune response) and the second corresponding to confirmed tumor progression (progression at TEP_interim_1, confirmed at TEP_interim_2, corresponding to true tumor escaping ICI treatments). To test this hypothesis, we compared the plasmatic concentration of free IL-18 at TEP 1 and observed that it is slightly higher (104 vs. 101) for the pseudo than the true progressors (Figure 2b). Despite being interesting, this result is not significant, and we cannot propose plasmatic levels of free IL-18 as a biomarker of ICI efficacy with the limited number of patients included in our cohort.

### 3.5. Levels of IL-18-Related Compounds and Their Association with Outcome of ICI-Treated NSCLC Patients

We next evaluated IL-18-related compound concentrations from NSCLC patients treated with ICIs and their association with outcomes using Kaplan–Meier analysis. Among all patients with available survival data and IL-18 concentrations available (N = 102), overall survival (OS) at 24 months was significantly linked to IL-18/IL-18BP complex concentration (concentration ≥ 237 pg/mL) (HR = 2.4; IC95% [1.4–4.1]; *p* = 0.00208, Figure 3b), and a trend was observed for free IL-18 (concentration ≥ 300 pg/mL) (HR = 1.6; IC95% [0.92–2.8]; *p* = 0.0918, Figure 3a).

We next evaluated the OS from ICI responding versus ICI nonresponding groups. No significant differences were observed within the responding group (N = 20) for both free IL-18 and IL-18/IL-18BP complex with *p* = 0.99 (Appendix A).

Among patients with initial tumor progression according to PERCIST criteria on PET1 (N = 69), OS at 24 months was significantly linked to IL-18/IL-18BP complex (concentration ≥ 237 pg/mL) with HR = 2.1 (IC95% [1.2–3.9]; *p* = 0.0155, as shown in Figure 3b). By contrast, no significant results were observed regarding the levels of free IL-18 in this subpopulation (Figure 3a). Furthermore, we observed no discernible impact of IL-18-related compound concentrations on progression-free survival (PFS) across all patient groups examined (Appendix A). 

These findings indicate that monitoring the plasma levels of the IL-18/IL-18BP complex is an efficient and easy way to stratify nonresponding patients. Patients with high levels of inactive IL-18 are linked to poor OS. By contrast, free IL-18 levels are not indicative of survival in this subgroup. 

### 3.6. Predictive Value of Combined IL-18-Related Compounds—Neutrophil Variables

The neutrophil-to-lymphocytes ratio, representing the balance between protumoral inflammatory status and antitumoral immune response, is recognized as a biomarker of response to ICIs [20]. In our cohort, no difference in NLR was evidenced between responding and nonresponding patients (spearman correlation, *p* = 0.8). We therefore focused on neutrophils to evaluate the predictive value at baseline of combined IL-18-related compounds and neutrophils. We first analyzed the predictive value of neutrophils alone.

Among all patients with survival data and neutrophils concentration available (N = 161), OS at 24 months was significantly linked to neutrophils concentration > 7.4 × 10^9^ neutrophils/L, with HR = 2.6 (IC95% [1.6–4.1]; *p* < 0.0001, as shown in Figure 4a, left panel). This difference is confirmed within the subgroup of nonresponding individuals (N = 101) with HR = 2.2 (IC95% [1.3–3.8]; *p* = 0.0036, Figure 4a, right panel). No significant differences were observed within the responding group, comprising a small number of patients (N = 40, *p* = 0.198, Appendix A). Furthermore, for PFS at 12 months, significant results were observed for all patients (N = 161) and, within the nonresponding group (N = 101), with, respectively, HR = 2.2 (IC95% [1.4–3.3]; *p* < 0.001, Figure 4b, left panel) and HR = 1.8 (IC95% [1.1–3]; *p* = 0.0117, Figure 4b, right panel). No significant difference was observed within the responding group (N = 46, *p* = 0.244, Appendix A).

To go further, multivariate analyses for OS at 24 months were first carried out within the entire cohort, including the cutoff of the IL-18/IL-18BP complex, free IL-18, and neutrophil concentration, as described in the Materials and Methods Section. We found that the IL-18/IL-18BP complex and neutrophil concentration fit well with OS, with, respectively, HR = 1.8 (IC95% = [1–3.2], *p* = 0.0458) and HR = 2.4 (IC95% = [1.3–4.5], *p* = 0.0044), and a trend was observed for free IL-18 (HR = 1.7, IC95% = [0.95–3.0], *p* = 0.0739). The final multivariate model allowed us to find a cutpoint of 0, associated with 72% sensibility and 62% specificity. Patients were distinguished according to whether they had a low or high risk of death. High-risk patients (HR = 2.5, CI95% = [1.4–4.5], *p* = 0.00226) were associated with shorter 24 months OS: 21% (CI95% = [11–39]), whereas low-risk patients had 52% (CI95% = [37–74]) (Figure 5a). These results indicate that combining high levels of IL-18/IL-18BP complex and neutrophil concentration efficiently identifies ICI-treated patients with poor OS. 

We performed the same analysis within the ICI nonresponding group and found that the cutoff of the IL-18/IL-18BP complex, free IL-18, and neutrophil concentration only showed a trend with OS, with, respectively, HR = 1.9 (IC95% = [1–3.7], *p* = 0.0599), HR = 1.7 (IC95% = [0.9–3.2], *p* = 0.1014), and HR = 1.8 (IC95% = [0.9–3.7], *p* = 0.94). The final multivariate model allowed us to find a cutpoint of 0, associated with 73% sensibility and 55% specificity. Patients were distinguished according to whether they had a low or high risk of death. High-risk patients (HR = 2.2, CI95% = [1.1–4.4], *p* = 0.018) were associated with shorter 24 months OS: 16% (CI95% = [7–35]), whereas low-risk patients had 49% (CI95% = [32–764]) (Figure 5b). Here, again, combining high levels of IL-18/IL-18BP complex and neutrophil concentration allows one to separate patients with better OS.

Together, these findings indicate that monitoring the plasma levels of inactive IL-18 and neutrophil concentration may enable one to stratify patients who stand to gain from alternative treatments, given their improved survival prognosis.

## 4. Discussion

To address the inefficacy of surgery for patients with advanced NSCLC, ICIs have become the standard treatment. However, only a limited subset of patients derive clinical benefits from these treatments [3]. The high cost and numerous immune-related adverse events of ICIs have prompted the scientific community to seek markers of effectiveness. For instance, immune-related genes and gene sets for predicting responses to anti-PD-1 therapy in NSCLC patients have been proposed [21]. Of interest, the immune-related gene set is composed of six genes and includes *IL18*. Further, an independent study highlighted the potential of IL-18 as a promising biomarker for anti-PD-1 treatment response in lung cancer [22]. In this study, the authors used a cytometric bead array immunoassay to show that the baseline plasma levels of IL-18 and CXCL10, over the 24 cytokines selected, were correlated with the degree of tumor response. In addition, a study conducted on the same cohort we utilized revealed a correlation between elevated levels of TNFα and CXCL2 and a positive response to ICI treatment [23]. TNFα being downstream of IL-18 signaling, this result motivated us to measure IL-18. 

In our study, we confirmed that circulating levels of IL-18 are increased in NSCLC patients versus healthy controls (Appendix A). This is likely due to inflammation linked to the tumor and/or to patients’ clinical history (such as previous chemotherapy or radiotherapy). The simultaneous increase in IL-18BP is not surprising since IL-18 levels are regulated by IL-18BP, and their expression levels are linked [6]. Recently, IL-18 BP was considered an immune checkpoint barrier by inhibiting IL-18-dependent antitumor immunity [10]. Further, knowing that the activity of IL-18 is balanced by IL-18BP, we determined the plasma levels of free IL-18, corresponding to active IL-18, together with the inactive form of IL-18, corresponding to the IL-18/IL-18BP complex. We show for the first time that NSCLC patients have lower levels of the inactive form of IL-18, complexed to IL-18BP, meaning that NSCLC patients would have higher levels of the active form of IL-18. The calculation of free IL-18 levels shows an increase in NSCLC compared with healthy subjects, yet it is not statistically significant. 

In addition, our investigation revealed a gradual decrease in the concentration of free IL-18 in patients undergoing ICI therapy. However, this effect is no longer observable when patients are categorized as responders versus nonresponders. Considering that ICI treatment enhances IFN-γ production [24], which in turn controls the production of IL-18BP [6], this could potentially account for the observed reduction in the concentration of free IL-18 in the plasma of patients treated with ICIs. However, in our cohort, the concentration of IL-18BP was not modulated by ICI treatment (Appendix A), and the concentration of IL-18 BP is not correlated to overall survival (*n* = 96 patients, HR = 1 (IC95 = [1-1], *p* = 0.98). Further, neither of the IL-18-related compounds was predictive of tumor response to ICIs, compared with the study by Wang and collaborators showing that the amount of total IL-18, among other cytokines, is predictive of ICI response [22]. Whether these differences were linked to ethnical origins, to patients’ clinical history (such as previous chemotherapy or radiotherapy), tumor stage, or to the method used to assay IL-18 remains to be determined. 

Classically, objective tumoral response is assessed by a Computed Tomography (CT) scan. Recent studies have shown that ^18^F-FDG PET, assessing changes in both tumor size and glucose metabolism, is also a valuable imaging modality to monitor tumor change with ICIs [25]. However, both CT and PET-CT imaging studies have limitations in the setting of ICI response evaluation. Indeed, they may depict an augmentation in tumor dimensions or the emergence of novel lesions, indicative of intratumor infiltration by immune cells, which do not correspond to treatment failure as these manifestations correlate with the efficacy of immunotherapy. Imaging studies of pseudoprogressor patients are indistinguishable from those of true progressors, justifying the identification of other reliable biomarkers to accurately distinguish between responding and nonresponding patients. Among predictive biomarkers, gene and epigenetic signatures, mutations and neoantigens, protein expression, and the presence of immune cells have been explored [26]. More recently, we evidenced the prognostic value of immunotherapy-induced organ inflammation assessed on interim ^18^FDG PET in advanced NSCLC patients [25]. Having shown in preclinical mouse models that active IL-18 sensitizes lung tumors to immunotherapy [9], we tested whether fluctuation of circulating levels of free IL-18 during the first week of ICI treatment may be a predictive biomarker for pseudoprogressors. Pseudoprogressors seem to have slightly higher levels of free IL-18 compared with true progressors. Although interesting, this difference is not significant, indicating the need to start a large-scale validation study. 

Furthermore, we found that circulating inactive IL-18 levels at baseline can discriminate NSCLC patients. More importantly, in patients with an early tumor increase assessed on ^18^FDG PET performed after 7 weeks of treatment, inactive IL-18 levels at baseline can discriminate between patients with different outcomes (Figure 3b). This new result is particularly valuable information as, at this step, the decision to stop or continue the treatment is difficult due to the possible pseudoprogression phenomenon. 

In patients with an initial PET progression but low inactive IL-18 levels, the ICI treatment could be continued, as their prognosis is more favorable than patients with high inactive IL-18 levels. By contrast, those with a concentration superior to the cut-off have a 2.3-fold increased risk of dying in 24 months. 

In addition, we showed that a low concentration of neutrophils at baseline is associated with good OS and PFS. This is true for all patients and ICI nonresponding patients [27]. We did not observe any correlation between the concentration of neutrophils and the levels of inactive IL-18 (Spearman correlation, *p* = 0.7), suggesting that these two variables are independent. It was previously reported in patients with resectable NSCLC that high intratumor CD66b-positive neutrophils correlated with a high incidence of relapse. More recently, the use of a high-plex digital spatial profiling approach demonstrated that CD66b is associated with resistance to ICI treatment in NSCLC [28]. Here, we showed that the concentration of circulating neutrophils discriminates both NSCLC patients and ICI refractory NSCLC patients with poor OS and PFS. Importantly, and for the first time, we were the first to describe a potent approach to discriminate ICI nonresponding patients with good OS. This approach, based on the determination of circulating levels of inactive IL-18 and neutrophils, is easily accessible, rapid, low cost, and reproducible. Indeed, neutrophil numeration is routinely performed during the clinical follow-up of patients, and measuring inactive IL-18 only requires one ELISA assay. 

There are limitations in this study. In particular, the ICI responding sample size was too small to allow the Cox model to function, which could explain why we were unable to conclude the prognostic value of free IL-18. In addition, we measured the levels of IL-18-related compounds in the plasma of patients, which could significantly differ from those found in the immediate vicinity of tumors, where immune cells directly interact with tumor cells and can participate in either pro or antitumor responses, as we recently discussed in [7]. Finally, some of the patients included in our cohort were previously treated with chemo or radiotherapy (see Table 1), which could also impact the levels of both IL-18-producing immune cells and IL-18-related compounds.

## 5. Conclusions

The bottleneck of ICI treatment is the identification of patients who will benefit from this therapy. We found that high plasma concentrations of inactive IL-18 in patients with increased lesions in TEP_interim_1 had significantly shorter overall survival probability. Moreover, when combined with neutrophil concentration, these two variables potently discriminate patients with good OS, which may rapidly guide the decision for a therapeutic intensification for ICI nonresponding patients. 

## Figures and Tables

**Figure 1 cancers-16-02226-f001:**
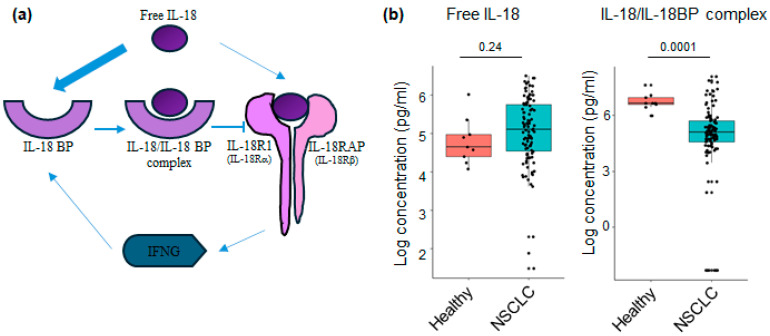
The concentration of IL-18-related compounds in NSCLC patients: (**a**) Mature free IL-18 binds with a high affinity to IL-18 binding protein (IL-18BP) to form an inactive IL-18/IL-18BP complex. It also binds to the IL-18 receptor, composed of IL-18R1 and IL-18RAP, to activate NF-kB, which in turn controls the production of IFN-γ. Finally, IFN-γ induces a negative feedback loop through the production of IL-18BP. (**b**) The concentrations of free IL-18 and IL-18/IL-18BP complex were measured in plasma samples before ICI treatment. The amount of biologically active, free IL-18 from patients and controls was calculated as described in the Materials and Methods Section. Bars show median levels. Each point represents one sample. *p*-values were determined by the Wilcoxon rank sum test with continuity correction.

**Figure 2 cancers-16-02226-f002:**
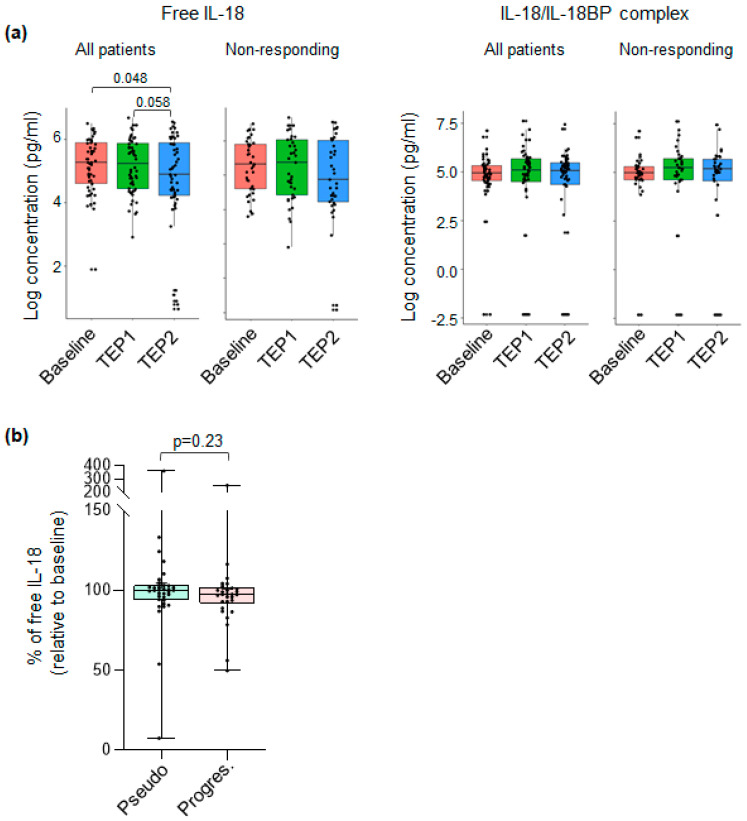
Free IL-18 concentration is decreased in ICI-treated patients: (**a**). Paired analysis of IL-18/IL-18BP complex and free IL-18 concentration at baseline, TEP1 and TEP2 from 123 individuals. *p*-values shown in Appendix A were determined by Wilcoxon test. Bars show median levels. Each point represents one sample. (**b**) Patients were divided into pseudoprogressors (light blue) or progressors (light pink) based on the TEP analysis, and IL-18 evolution between baseline and TEP1 was expressed as the percent of IL-18 at TEP1 relative to baseline. Pseudoprogressor *n* = 34, progressor *n* = 30. Bars show median levels. Each point represents one sample. *p*-values were determined by Mann–Whitney test.

**Figure 3 cancers-16-02226-f003:**
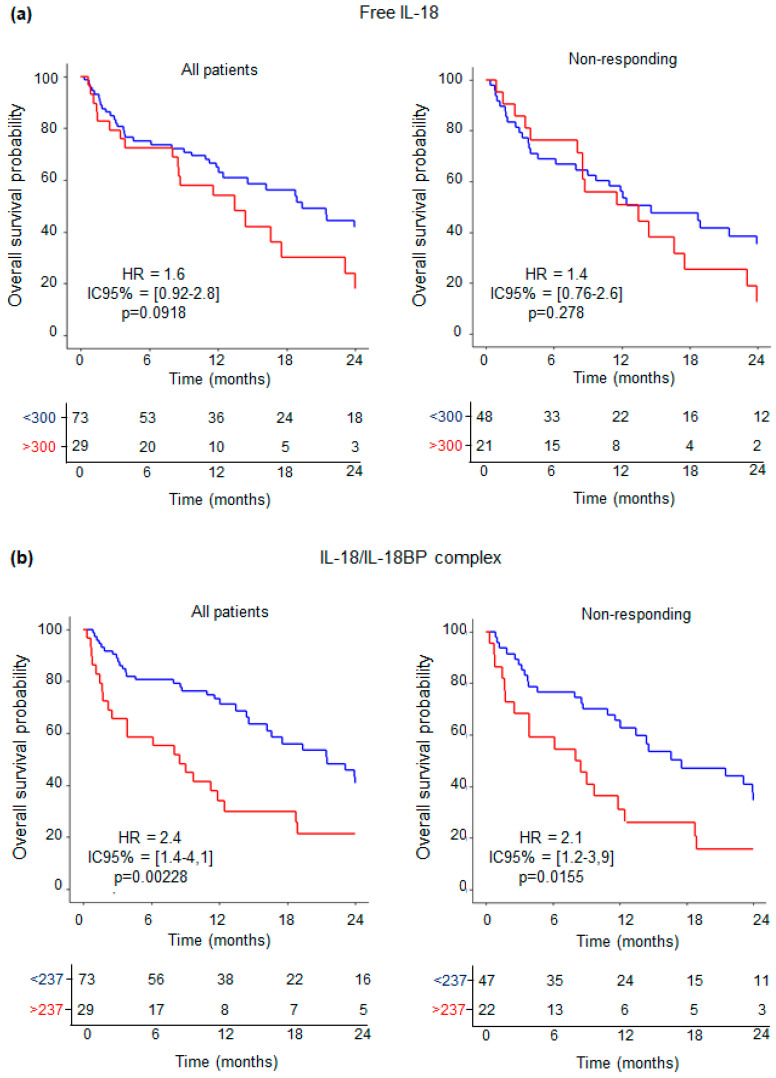
Overall survival is increased in ICI nonresponding patients with low free IL-18. Survival analyses (Kaplan–Meier) of NSCLC patients showing all patients (left) and ICI nonresponding patients (right): (**a**) Free IL-18 and (**b**) IL-18/IL-18 complex. Blue lines correspond to concentrations lower than the cutoff. Red lines correspond to concentrations higher than the cutoff. Numbers at risk with censored data are shown. *p*-values were determined by Mentel–Cox test. HR: hazard ratio; IC: interval confidence.

**Figure 4 cancers-16-02226-f004:**
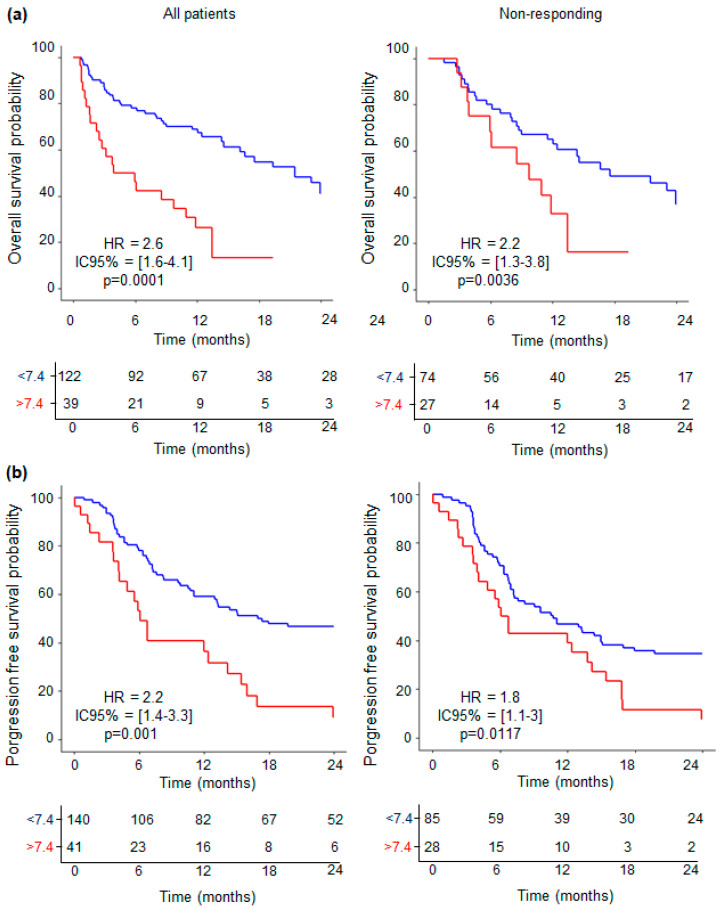
Overall survival is increased in ICI nonresponding patients with low neutrophil concentration. Survival analyses (Kaplan–Meier): (**a**) Overall survival and (**b**) progression-free survival. Blue lines correspond to concentrations lower than the cutoff. Red lines correspond to concentrations higher than the cutoff. Numbers at risk with censored data are shown. *p*-values were determined by Mentel–Cox test. HR: hazard ratio; IC: interval confidence.

**Figure 5 cancers-16-02226-f005:**
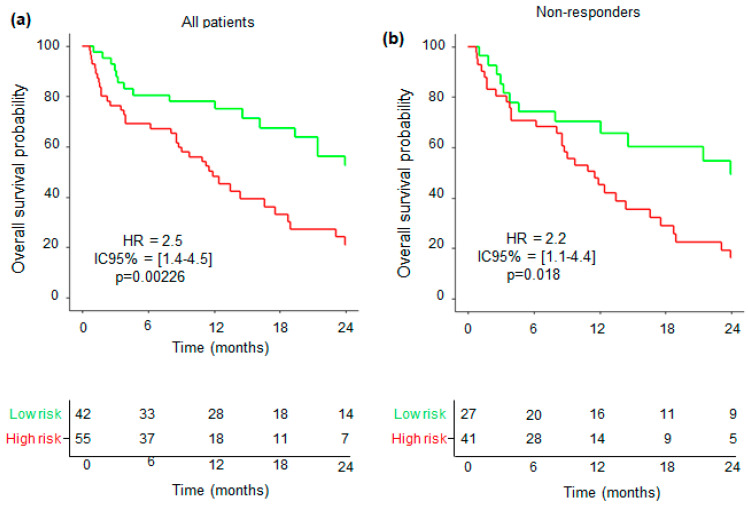
Overall survival of patients with combined prognostic factors. Survival analyses (Kaplan–Meier): (**a**) All patients and (**b**) nonresponding patients. Green lines correspond to patients with low risk (3 selected variables are lower than the cutoff) and red lines correspond to high risk (3 selected variables are higher than the cutoff). Numbers at risk with censored data are shown. *p*-values were determined by Mentel–Cox test.

**Table 1 cancers-16-02226-t001:** Main epidemiological data.

Characteristics		*n*, (Range) or (%)
Age (years)	Median (range)	66 (39–91)
Sex, *n*, (%)	Female	88 (36.8%)
	Male	151 (63.2%)
Smoking history, *n*, (%)	Smokers or smoking history	221 (84%)
	Nonsmoker	42 (16%)
Body Mass Index WHO categories	Underweight (<18.5)	26 (15.3%)
	Normal weight (18.5 to 24.9)	88 (51.8%)
	Overweight (25 to 30)	45 (26.5%)
	Obese (≥30)	11 (6.5%)
	Unknown	102 (37.5%)
Tumor histology, *n*, (%)	Undifferentiated carcinoma	11 (4.3%)
	Adenocarcinoma	200 (77.5%)
	Squamous cell carcinoma	47 (18.2%)
Tumor stage	IIIB	23 (15%)
	IV	130 (85%)
Immunotherapy, *n*, (%)	Nivolumab	29 (20.8%)
	Pembrolizumab	104 (74.8%)
	Atezolizumab	6 (4.3%)
PD-L1 tumor expression, *n* (%)	>50%	88 (51.5%)
	0%	28 (16.4%)
	1–49%	55 (32.2%)
Previous lung surgery, *n*, (%)	No	84 (61.3)
	Yes	53 (19.4)
Previous radiotherapy, *n* (%)	No	84 (61.3%)
	Yes	53 (38.7%)
Previous chemotherapy lines	No	11 (4.04%)
	1st line	88 (33.7%)
	2nd line	83 (31.8%)
	3rd line	90 (34.5%)
PERCIST response	Complete response	12 (5%)
	Partial response	55 (23.1%)
	Progressive disease	66 (60%)
	Stable disease	18 (7.6%)
	Unknown	34 (12.5%)
Pseudo progression	No	106 (48%)
	Yes	115 (52%)

## Data Availability

The data presented in this study are available in this article and Appendix A.

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
