# Peer review of "Plasmatic Inactive IL-18 Predicts a Worse Overall Survival for Advanced Non-Small-Cell Lung Cancer with Early Metabolic Progression after Immunotherapy Initiation"

_cancers, 2024, doi:10.3390/cancers16122226_

Round 1

Reviewer 1 Report

Comments and Suggestions for Authors

The authors present an interesting analysis exploring the potential use of free IL-18 levels as a biomarker for overall survival in advanced NSCLC patients treated with immunotherapy. However, there are several statistical issues that need to be addressed before the conclusions can be considered valid.

Major Comments:

Figure 1b:

The data for IL-18 complex levels appears to have unequal variances between groups, violating the assumption of homogeneity of variance required for the Student's t-test. Additionally, the authors should check the distribution of the data and consider using a non-parametric test if the data is not normally distributed.

Cutoff determination for survival analysis (Section 3.5)

The authors have not provided a clear explanation for how the cutoff value for IL-18 complex levels was determined for the survival analysis. If multiple cutoff values were tested to find a statistically significant result, this would significantly increases the risk of false positive findings. The authors should clarify their method for determining the cutoff value.

Author Response

During the review, we identified an error in the calculation of free IL-18 concentrations. The revised version of the paper includes the corrected values. Notably, the new results strengthen our conclusions and underscore the importance of monitoring plasmatic concentrations of inactive IL-18 alongside neutrophils in NSCLC patients treated with immune checkpoint inhibitors (ICIs). This approach may help distinguish ICI non-responding patients with better overall survival (OS), potentially guiding rapid decisions for therapeutic intensification.

These new results motivated our decision to change the title of the manuscript.

Comments and Suggestions for Authors.

The authors present an interesting analysis exploring the potential use of free IL-18 levels as a biomarker for overall survival in advanced NSCLC patients treated with immunotherapy. However, there are several statistical issues that need to be addressed before the conclusions can be considered valid.

Major Comments:

Figure 1b:

The data for IL-18 complex levels appears to have unequal variances between groups, violating the assumption of homogeneity of variance required for the Student's t-test. Additionally, the authors should check the distribution of the data and consider using a non-parametric test if the data is not normally distributed

We thank the reviewer for this comment. The tests were presented as two-tailed unpaired t-tests, while Wilcoxon rank sum tests were performed. The legend to Figure 1b has been modified accordingly (l. 266).

Cutoff determination for survival analysis (Section 3.5)

The authors have not provided a clear explanation for how the cutoff value for IL-18 complex levels was determined for the survival analysis. If multiple cutoff values were tested to find a statistically significant result, this would significantly increases the risk of false positive findings. The authors should clarify their method for determining the cutoff value.

We did not include threshold determination in the statistical method. The statistical analysis was modified accordingly. l. 215-225.

The thresholds were determined by brute force, and the patients were divided into high-risk and low-risk groups on the basis of a logrank test. They were sorted according to p-value. The lowest p-value was retained

However, only the thresholds presented in the article were tested. There was only one test per IL-18 related compound. Consequently, we did not increase the risk of false positive results. Only a few variables were significant in the univariate analysis, so we did not correct the p-value for multiple testing.

Reviewer 2 Report

Comments and Suggestions for Authors

An interesting article that will undoubtedly attract the attention of readers is presented for review.

1. Inclusion criteria are stage IIIB or IV NSCLC, however, Table 1 shows others stages, starting with I. Did the authors mean the tumor size in the table (T)? Provide correct staging, including how many stage IIIB and IV patients should be included. Provide information about the involvement of lymph nodes and the location of metastases.

2. What does Unknown mean in the Immunotherapy line? Are these patients then not included in the review? Why are they in the table then?

3. The study group is extremely heterogeneous. Why did the authors not perform a multivariate survival analysis? How to understand what has a stronger effect on survival rates - free IL-18 or the fact that the patient did not undergo surgery? Is this his first or third line of therapy? Are metastases localized, for example, in the brain or liver? Without multivariate analysis, the results are difficult to interpret correctly. I recommend that the authors carry out additional calculations.

Author Response

During the review, we identified an error in the calculation of free IL-18 concentrations. The revised version of the paper includes the corrected values. Notably, the new results strengthen our conclusions and underscore the importance of monitoring plasmatic concentrations of inactive IL-18 alongside neutrophils in NSCLC patients treated with immune checkpoint inhibitors (ICIs). This approach may help distinguish ICI non-responding patients with better overall survival (OS), potentially guiding rapid decisions for therapeutic intensification.

These new results motivated our decision to change the title of the manuscript.

Comments and Suggestions for Authors

An interesting article that will undoubtedly attract the attention of readers is presented for review.

  1. Inclusion criteria are stage IIIB or IV NSCLC, however, Table 1 shows others stages, starting with I. Did the authors mean the tumor size in the table (T)? Provide correct staging, including how many stage IIIB and IV patients should be included. Provide information about the involvement of lymph nodes and the location of metastases.

The UICC stages indicated correspond to the discovery of NSCLC disease, which may indeed be confusing for the reader. At the time of ICI treatment, all patients were stage IIIB or IV. This has been corrected in Table 1.

The patient characteristics listed in Table 1 were derived from an ancillary study, wherein patients from the ECMI and IMMUN-FDG cohorts were prospectively enrolled. At the time of their inclusion in 2019, only the UICC classification was requisite for patient enrollment. Notably, neither lymph node involvement nor the location of metastases were specified. 

  1. What does Unknown mean in the Immunotherapy line? Are these patients then not included in the review? Why are they in the table then?

We thank the reviewer for this comment. This was an error, this line has been deleted.

  1. The study group is extremely heterogeneous. Why did the authors not perform a multivariate survival analysis? How to understand what has a stronger effect on survival rates - free IL-18 or the fact that the patient did not undergo surgery? Is this his first or third line of therapy? Are metastases localized, for example, in the brain or liver? Without multivariate analysis, the results are difficult to interpret correctly. I recommend that the authors carry out additional calculations

The relationship between patient characteristics and OS was studied. However, only two variables were statistically significant:

  • Tumor histology, with p<0.001 for OTHER. But with an enrolment of five, we decided that the results were due to a small population.
  • Pseudo progression with p=0.018. However, as pseudo progression is unknown for 66 patients, the multivariate modelling, initially based on 93 patients (who have the data), falls to 63 patients, losing 1/3 of the patients. Furthermore, pseudo progression is a prognostic variable determined after baseline. It is therefore of less interest than baseline levels of Il18 for predicting prognosis because it can only be used at a later stage.

Reviewer 3 Report

Comments and Suggestions for Authors

Comments:

1. Please provide all the abbreviations.

2. On Table 1: need more patients' info. such as HbA1c, BUN, and creatinine levels. Also please add BMI, alcohol history, family history, radiation history, asbestos, arsenic, and diesel exposure history, radon gas expsoure history, etc.

3. On Figure 1: please explain why the sample size of healthy patients is small.

4. IL-1 is closely related to IL18. Do you have IL-1 data? If yes, what would be the IL-1 results and conclusion?

5. TNF is one of the IL-18 downstream gene. Do you have TNF data? If yes, what would be the TNF results and conclusion?

6. IL-18 binds to IL-18 receptor. What is IL-18 receptor role in this study?

7. I recommend to use full name of NSCLC in the title of this study.

Author Response

During the review, we identified an error in the calculation of free IL-18 concentrations. The revised version of the paper includes the corrected values. Notably, the new results strengthen our conclusions and underscore the importance of monitoring plasmatic concentrations of inactive IL-18 alongside neutrophils in NSCLC patients treated with immune checkpoint inhibitors (ICIs). This approach may help distinguish ICI non-responding patients with better overall survival (OS), potentially guiding rapid decisions for therapeutic intensification.

These new results motivated our decision to change the title of the manuscript.

Comments and Suggestions for Authors

Comments:

  1. Please provide all the abbreviations.

We added an abbreviation list (l. 524) in the revised manuscript.

  1. On Table 1: need more patients' info. such as HbA1c, BUN, and creatinine levels. Also please add BMI, alcohol history, family history, radiation history, asbestos, arsenic, and diesel exposure history, radon gas expsoure history, etc.

The patient characteristics listed in Table 1 were derived from an ancillary study, wherein patients from ECMI and IMMUN-FDG cohorts were prospectively enrolled. At the time of their inclusion in 2019, only the UICC classification was requisite for patient enrollment. Therefore, only standard biological and clinical data were provided. These data did not include HbA1c, BUN and creatinine levels nor alcohol history, family history, radiation history, asbestos, arsenic, and diesel or radon gaz exposure history.

However, we have access to the BMI and we now included it in the Table 1. The relationship between BMI, OS, and IL-18 related compounds was studied. BMI is significant in univariate analysis. We grouped it into two categories (normal+underweight vs. overweight+obese) to increase sample size and because the statistical behavior of the subgroups was similar. This new variable, BMI_R, is statistically significant in both univariate and multivariate analyses. However, it reduces the quality of the model, making IL-18 levels less significant or not significant at all, and the predictive score less accurate. Therefore, we believe this variable should not be considered.

  1. On Figure 1: please explain why the sample size of healthy patients is small.

We thought it would be interesting to compare the levels of IL-18 in healthy and NSCLC patients. When we asked to the biobank, only few healthy samples were available and we included all of them.

  1. IL-1 is closely related to IL18. Do you have IL-1 data? If yes, what would be the IL-1 results and conclusion?

We agree with the reviewer IL-1β is closely related to IL-18, although it's worth noting that while pro-IL-1 mRNA requires a priming step for production, pro-IL-18 mRNA is constitutively expressed.  We previously showed in lung tumor mouse models that IL-18, and not IL-1 β, sensitized lung tumor to immunotherapy (PMID 33510147). We hypothesized that in humans, IL-18 may act in a similar manner. If so, IL-18 levels could predict the efficacy of ICI in NSCLC. Therefore, we only measured IL-18 levels in this study.

  1. TNF is one of the IL-18 downstream gene. Do you have TNF data? If yes, what would be the TNF results and conclusion?

Our team has published an association between high levels of TNFa and CXCL2 with a favorable response to ICI (PMID: 35994416). As mentioned by the reviewer, TNFa being downstream of IL-18 signaling, this result was one of the arguments that prompted us to measure IL-18. This is now addressed in the discussion (l. 435).

  1. IL-18 binds to IL-18 receptor. What is IL-18 receptor role in this study?

Free IL-18 binds to IL-18 receptor (composed of the IL-18R1 subunit recruiting the high affinity IL-18RAP subunit for MYD88/NF-κB activation). In this study, we focused on free IL-18, corresponding to the active form. It is worth noting that only a few studies have addressed the activity of IL-18, i.e., the free, unbound IL-18 from IL-18BP, as we previously discussed in PMID: 37298187.

Regarding IL-18R, the full receptor expression (IL-18R1 and IL-18RAP) was reported to be low in peripheral blood from NSCLC (PMID: 28811967), whereas its expression increases in a small fraction of CD8+ T cells (T-bet+Eomes+) in tumor tissue. Of importance, this receptor expression is linked to the production of IFN-g and therefore marks functional CD8+ T cells in untreated NSCLC. To our knowledge no study described the feature of IL-18R in ICI-treated NSCLC patients.

  1. I recommend to use full name of NSCLC in the title of this study.

We thank you for this remark and we now add the full name in the title.

Reviewer 4 Report

Comments and Suggestions for Authors

This study aimed to investigate the role of IL-18 and IL-18BP in predicting treatment response in advanced NSCLC patients undergoing ICI therapy. It demonstrated elevated levels of circulating IL-18 and IL-18BP in NSCLC patients, suggesting that IL-18BP binds to active IL-18, potentially affecting treatment efficacy. However, these results should be validated by additional studies to have clinical implications. Here are a few comments for the author:

Introduction:

1. Provide a detailed explanation of the connection between IL-18 and immunotherapy in NSCLC.

2. Add more background information on the PERCIST Criteria and its relevance to the study design.

Ensure that the presentation on IL-18 and tumor progression is clear and coherent, highlighting the key findings and their implications for the study.

Methods:

4. Some sections of this part are overly descriptive. For example, the description of exclusion criteria could be more concise and reader-friendly. Instead of listing specific types of exclusion criteria (e.g., "histological subtype other than adenocarcinoma, squamous cell carcinoma, or undifferentiated carcinoma"), it could be summarized more succinctly.

5. Although this section provides an overview of the study, it needs to be more specific in certain areas. For instance, clarify the amount of blood drawn, the methods used for sample collection and processing, and any quality control measures taken.

6. Various studies, protocols, and criteria are mentioned throughout the text (e.g., "PERCIST criteria" or "Article L1121-5 to -8 of the French Public Health Code"), but they are not explained in detail. Providing a brief explanation or reference for such terms would be helpful, especially for readers who are not familiar with the same specialization or field of study.

Results:

7. Discuss limitations, such as the retrospective nature of the analysis, confounding variables, and the need for validation in larger cohorts.

Discussion:

8. While the study refers to IL-18BP, which limits the amount of active IL-18, it would be insightful to explain or reference studies that explore how this balance modulates treatment response. The section briefly mentions the association of neutrophil concentration with treatment response but fails to elaborate on the potential implications and underlying mechanisms of this relationship. Expanding on this point would make it clearer.

Author Response

During the review, we identified an error in the calculation of free IL-18 concentrations. The revised version of the paper includes the corrected values. Notably, the new results strengthen our conclusions and underscore the importance of monitoring plasmatic concentrations of inactive IL-18 alongside neutrophils in NSCLC patients treated with immune checkpoint inhibitors (ICIs). This approach may help distinguish ICI non-responding patients with better overall survival (OS), potentially guiding rapid decisions for therapeutic intensification.

These new results motivated our decision to change the title of the manuscript.

Comments and Suggestions for Authors

This study aimed to investigate the role of IL-18 and IL-18BP in predicting treatment response in advanced NSCLC patients undergoing ICI therapy. It demonstrated elevated levels of circulating IL-18 and IL-18BP in NSCLC patients, suggesting that IL-18BP binds to active IL-18, potentially affecting treatment efficacy. However, these results should be validated by additional studies to have clinical implications. Here are a few comments for the author:

We thank the reviewer for its valuable work.

In this study we aim to investigate whether plasmatic levels of IL-18 may predict ICI response in advanced NSCLC. What is important to note, is that plasmatic IL-18 comprises both IL-18 bound to IL-18 Binding Protein (IL-18BP), an inactive IL-18 related compound named IL-18BP/IL-18 complex in the manuscript, and free IL-18, corresponding to the active form which binds to IL-18R. Further, only a few studies have addressed the activity of active IL-18, i.e., the free, unbound IL-18 and none of them were performed on ICI-treated NSCLC patients.

To conduct our study, we mainly focused on the levels of active free IL-18 and inactive IL-18/IL-18BP complex. We showed that high plasma concentration of inactive complexed IL-18 in patients with increased lesions in TEPinterim1 had significantly shorter overall survival probability. Moreover, when combined to neutrophil concentration, these two variables potently discriminate patients with good OS, which may facilitate decision to a therapeutic intensification for ICI non-responding patients.

Introduction:

  1. Provide a detailed explanation of the connection between IL-18 and immunotherapy in NSCLC.

We thank the reviewer for this comment highlighting the lack of clarity in our message.

We now give more information in the introduction (l. 60 to 75).

  1. Add more background information on the PERCIST Criteria and its relevance to the study design.

In the present study, which aimed to identify new predictive biomarkers of immunotherapy efficacy, 18FDG PET/CT was the imaging modality chosen to monitor treatment response. Our team has previously published results highlighting the relevance of PET/CT in ICI-treated lung cancer. We have shown that 18FDG PET enables early assessment of treatment-induced metabolic changes in the tumor or tissue inflammation linked to immune-related adverse events (PMID: 38649279; PMID: 35562529; PMID: 31760467). We also participated in the drafting of guidelines/procedural standards on the recommended use of 18FDG PET/CT imaging during immunomodulatory treatments in patients with solid tumors (PMID: 35376991).

When using PET/CT exams, the standard criteria to evaluate tumor response are the “Positron Emission Tomography Response Criteria in Solid Tumors” (PERCIST) defined by Wahl et al. in 2009 (PMID 19403881). The criteria consist of four categories: complete metabolic response (CMR), partial metabolic response (PMR), progressive metabolic disease (PMD), and stable metabolic disease (SMD).

Complete metabolic response (CMR) is defined as the complete resolution of 18F-FDG uptake within the measurable target lesion.

Partial metabolic response (PMR) is defined by 1) the reduction of a minimum of 30% in target measurable tumor 18F-FDG SUL peak, with absolute drop in SUL of at least 0.8 SUL units 2) no increase >30% of SUL or size in all other lesions 3) no new lesions. SUL is the standardized uptake value corrected for lean body mass. SULpeak (or SULpeak) is the peak SUL in a spherical 1 cm3 volume of interest (VOI).

Progressive metabolic disease (PMD) is defined as a >30% increase in 18F-FDG SUL peak, with >0.8 SUL units increase in tumor SUL from the baseline scan, or visible increase in the extent of 18F-FDG tumor uptake, or new 18F-FDG

Stable metabolic disease (SMD) is defined as no CMR, PMR and no PMD.

We have modified the introduction accordingly (l. 94 to 109).

Ensure that the presentation on IL-18 and tumor progression is clear and coherent, highlighting the key findings and their implications for the study.

We previously showed in lung tumor mouse models that IL-18, and not IL-1 β, sensitized lung tumor to immunotherapy (PMID 33510147). We hypothesized that in humans, IL-18 may act in a similar manner. If so, IL-18 levels could predict the efficacy of ICI in NSCLC. This is what we tested in this study. As stated earlier, we have modified the introduction to enhance its clarity and understandability.

Methods:

  1. Some sections of this part are overly descriptive. For example, the description of exclusion criteria could be more concise and reader-friendly. Instead of listing specific types of exclusion criteria (e.g., "histological subtype other than adenocarcinoma, squamous cell carcinoma, or undifferentiated carcinoma"), it could be summarized more succinctly.

We thank the reviewer for this comment. The inclusion criteria were the following: (1) pathologically proven stage IIIB or IV NSCLC; (2) an indication to start ICI in monotherapy and in first or later line; (3) ECOG performance status of 0 to 2; (4) age of at least 18 years (5) histological corresponding to adenocarcinoma, squamous cell carcinoma or undifferentiated carcinoma; (6) no patient opposition (7) no measurable lesion by PERCIST V1.0. We have changed the text accordingly (l. 139 to 148).

  1. Although this section provides an overview of the study, it needs to be more specific in certain areas. For instance, clarify the amount of blood drawn, the methods used for sample collection and processing, and any quality control measures taken.

Two EDTA blood tubes (BD vacutainer, 10 ml) were drawn from each patient. Each tube was centrifuged at 815g for 10 min at 4°C and the collected supernatant was centrifuged at 2500g for 10 min at 4°C. Plasma was aliquoted (1ml/aliquot) and stored at -80°C until use. Tubes were store in accredited biobank (ISO 9001, NFS 96-900).

We added this information in the revised version of the manuscript (l. 162 to 167).

  1. Various studies, protocols, and criteria are mentioned throughout the text (e.g., "PERCIST criteria" or "Article L1121-5 to -8 of the French Public Health Code"), but they are not explained in detail. Providing a brief explanation or reference for such terms would be helpful, especially for readers who are not familiar with the same specialization or field of study.

We have provided a clear explanation of the PERCIST criteria (l. 97 to 105). To facilitate the readability of our manuscript, we have removed the reference from the French Public Health Code.

Results:

  1. Discuss limitations, such as the retrospective nature of the analysis, confounding variables, and the need for validation in larger cohorts.

Following the recommendation of MDPI and the Cancers’ Template, we have discussed these limitations in the discussion (l. 510 to 518).

Discussion:

  1. While the study refers to IL-18BP, which limits the amount of active IL-18, it would be insightful to explain or reference studies that explore how this balance modulates treatment response.

We thank the reviewer for this comment. Indeed, IL-18 BP has been described to function as an immune checkpoint barrier to IL-18 immunotherapy in mouse preclinical models (ref. 8 in our manuscript). Whereas IL18 and CXCL10 protein levels are indicative of ICI response in lung cancer patients, IL-18 BP was not mentioned in this study (ref 23 in our manuscript). To our knowledge we are the first to study IL-18 BP protein levels in patients under ICI treatment. Our data showed that IL-18BP protein contents is not significantly modulated in non-responding and responding patient and during the treatment (See Supplementary Table 1 and 2). Further, the concentration of IL-18 BP is not correlated to overall survival (n=96 patients, HR =1 (IC95 = [1-1], p=0.98. These results were discussed in the manuscript (l. 452 to 464) and we modified the discussion in light of our findings.  We do hope that this revised version is now suitable for publication.

The section briefly mentions the association of neutrophil concentration with treatment response but fails to elaborate on the potential implications and underlying mechanisms of this relationship. Expanding on this point would make it clearer.

We thank the reviewer for this comment. We now gave more explanation on the potential implications of neutrophils and IL-18 related compounds in the discussion (see l 495-509).

Round 2

Reviewer 1 Report

Comments and Suggestions for Authors

There appears to be a discrepancy between the information presented in Figure 1b and Figure 3b. In Figure 1b, which shows a box plot, the log value of approximately 2.47 (log300) has very few data points or dots visible below that level on the y-axis. However, Figure 3b, indicates that there were 73 patients with a log value under 2.47 (as shown in Figure 1b). This contradiction raises questions about the consistency and accuracy of the data representation across these two figures, and further clarification or explanation might be needed to reconcile this apparent mismatch.

Author Response

We thank the reviewer for his/her comments highlighting the lack of clarity of our manuscript.

Figure 1b presents a box plot displaying the concentration levels of free IL-18 and the IL-18/IL-18BP complex for each NSCLC patient in our cohorts, as well as for healthy donors (refer to lines 263-266). As described in the Materials and Methods section, the data were normalized using the basic logarithmic function (line 206). Accordingly, the Y-axis legend is labeled as Log concentration (pg/ml).

In Figure 3b, the Kaplan-Meier analysis illustrates the survival outcomes of all NSCLC patients (left) and ICI non-responding NSCLC patients (right). This analysis includes a total of 102 patients: 73 patients with IL-18/IL-18BP complex concentrations below the cutoff and 29 patients with concentrations above it. The cutoff value is 237 pg/ml. The method for calculating this cutoff is detailed in lines 216-226. To enhance clarity, we have updated the legend of Figure 3 (lines 328-329).

Reviewer 2 Report

Comments and Suggestions for Authors

The authors significantly revised the manuscript and corrected shortcomings. I have no further comments on the manuscript.

Author Response

We thank the reviewer for his/her positive comments

Reviewer 3 Report

Comments and Suggestions for Authors

The authors answered my comments and questions. No more comments.

Author Response

We thank the reviewer for this comment

Reviewer 4 Report

Comments and Suggestions for Authors

No further comments 

Author Response

We thank the reviewer for this comment

Round 3

Reviewer 1 Report

Comments and Suggestions for Authors

Thanks for the update.